# Identify Critical Nodes in Complex Network with Large Language Models

## Abstract

Identifying critical nodes in networks is a classical combinatorial optimization task, and many methods struggle to strike a balance between adaptability and utility. Therefore, we propose an approach that empowers Evolutionary Algorithm (EA) with Large Language Models (LLMs), to generate a function called "score_nodes" which can further be used to identify crucial nodes based on their assigned scores. Our model consists of three main components: Manual Initialization, Population Management, and LLMs-based Evolution, and it evolves from initial populations with a set of designed node scoring functions created manually. LLMs leverage their strong contextual understanding and rich programming techniques to perform crossover and mutation operations on the individuals, generating new functions. These functions are then categorized, ranked, and eliminated to ensure the stable development of the populations while preserving diversity. Extensive experiments demonstrate the excellent performance of our method compared to other state-of-the-art algorithms. It can generate diverse and efficient node scoring functions to identify critical nodes in the network.

## 1 Introduction

Various networks widely exist in the real world, such as social networks, biological networks, etc West et al. (2001). For the network, identifying an optimal set of critical nodes for which the activation/removal has the maximized impact is a basic research problem with broad applications in epidemic control, influence maximization, etc Lalou et al. (2018); Arulselvan et al. (2009). From the perspective of combinatorial optimization, this problem could be considered as a kind of optimal search task, in which the computational complexity increases exponentially along with the size of the network Taniguchi et al. (2006). Existing approaches to this problem can be categorized as follows: from heuristic methods to optimization-based methods, and finally, to learning-based methods.

The first kind of work is the basic methods based on heuristics, such as using degree centrality Zhang and Luo (2017) to judge whether the node is critical or not, which is simple yet very effective and efficient, showing superior performance in many kinds of networks Arulselvan et al. (2009); Mao et al. (2023). The second kind of work further tries to re-formulate the problem with approximated optimization problems. Some other algorithms like CoreHD Zdeborová et al. (2016) and WeakNeighbors Schmidt et al. (2019) utilize $k$-core decomposition, in which the $k$-core of a network is defined as the maximal subgraph with every node having a degree of at least $k$. These methods progressively eliminate nodes with the highest degrees within their respective $k$-cores to identify key nodes. Furthermore, Min-Sum Braunstein et al. (2016) focuses on creating an acyclic network by removing nodes that form loops, followed by utilizing greedy tree-breaking algorithms to decompose the remaining forest into smaller disconnected components. Another work, GND Ren et al. (2019), adopts a spectral approach to iteratively partition the network into two components. These methods have the advantage of fast computation, but they may lead to sub-optimal solutions, or even lack a suitable solution altogether. Recently, with the advancement of deep learning, some works Zhang and Wang (2022); Grassia et al. (2021) approach the problem by using neural networks to learn the node importance or ranking nodes adaptively. That is, they utilize neural networks to assign scores to network nodes, determining their importance based on these scores.

Despite these advances, there still exists a key challenge of *adaptability-utility dilemma*, which largely affects the real application values of these methods. Specifically, the heuristic or optimization

methods provide the same solution for all kinds of networks, but ignoring the different characteristics, which leads to poor performance in those unpopular networks. The learning-based methods, on the other hand, try to search the critical nodes adaptively, but are faced with high search costs in such a huge search space, which in turn, leads to poor utility.

Recently, Large Language Models (LLMs) have shown a human-like ability in many tasks, from basic natural language processing, to code generation, and some high-level logical reasoning and decision-making tasks Zhao et al. (2023); Chang et al. (2023); Gao et al. (2023). Inspired by these advances, we propose a novel solution that adopts LLMs to generate the solution and design an evolutional learning framework to fully unleash LLMs' power. Specifically, we utilize LLMs-empowered evolutionary algorithms to address the critical node problem, with LLMs performing crossover and mutation operations, continuously iterating to generate new solutions.

The contributions of this work can be summarized as follows:

- We approach the fundamental problem of critical node detection from a brand new perspective of LLMs, for which the original problem is transformed into a code-generation task.

- We propose a general framework integrating LLMs and evolutional learning, and a generation-update mechanism for LLMs and the crossover-mutation method for evolutional learning.

- We conduct extensive experiments on both real-world and synthetic networks, and the results show the superior performance of our proposed method. The discoverd solutions can further inspire researchers and practitioners in this area.

## 2 PROBLEM STATEMENT

Given a network $\mathcal{G} = (\mathcal{V}, \mathcal{E})$, where $\mathcal{V}$ represents the set of nodes $\{v_1, v_2, ..., v_{|\mathcal{V}|}\}$ and $\mathcal{E}$ represents the set of edges $\{e_1, e_2, ..., e_{|\mathcal{E}|}\}$, our objective is to identify a sequence of critical nodes $(v_1, v_2, \cdots, v_N)$, whose deletion makes the network disconnected Lalou et al. (2018). These critical nodes will be obtained through a function $f$ generated by our method, evaluating the importance of the nodes for the network.

For the sake of facilitating function processing, we represent the network $\mathcal{G}$ as an adjacency matrix $\mathbf{A} = \{0, 1\}^{|\mathcal{V}| \times |\mathcal{V}|}$, where $\mathbf{A}_{jk} = 1$ indicates that there is an edge between node $v_j$ and node $v_k$; otherwise, $\mathbf{A}_{jk} = 0$. Many existing methods rely on a crucial parameter, the number of target nodes $N$, directly impacting the algorithm's performance. To ensure the efficiency and generalization of the function learned by our approach, our objective node scoring function $f$ does not include the variable $N$. Instead, it will take the network's adjacency matrix $\mathbf{A}$ as input and produce scores for each node $\mathcal{S} = \{s_{v_1}, s_{v_2}, ..., s_{v_{|\mathcal{V}|}}\}$ as output. In other words, the formulation $\mathcal{S} = f(\mathbf{A})$ enables us to rank nodes based on their scores and obtain any desired quantity of critical nodes.

## 3 METHODOLOGY

In order to discover an optimal node scoring function $f$, our model leverages LLMs with their powerful contextual understanding and programming capabilities, within an evolutionary learning framework to fully unleash the potential of LLMs. Through generation, evolution, evaluation, and iteration, we can obtain a novel high-performance function. The model primarily consists of three components: Manual Initialization, LLMs-based Evolution, and Population Management. Figure 1 illustrates the framework of our model.

### 3.1 MANUAL INITIALIZATION

Evolutionary algorithms are a class of population-based stochastic search strategies, and the initial population provides potential individuals for the entire search process. These initial individuals continually iterate and improve during the training process, getting closer to the optimal solution. A well-designed initialization can assist evolutionary algorithms in finding the optimal solution, particularly when dealing with large-scale optimization problems using finite-sized populations. Therefore, we categorize the design approaches for node scoring functions into two classes as follows.

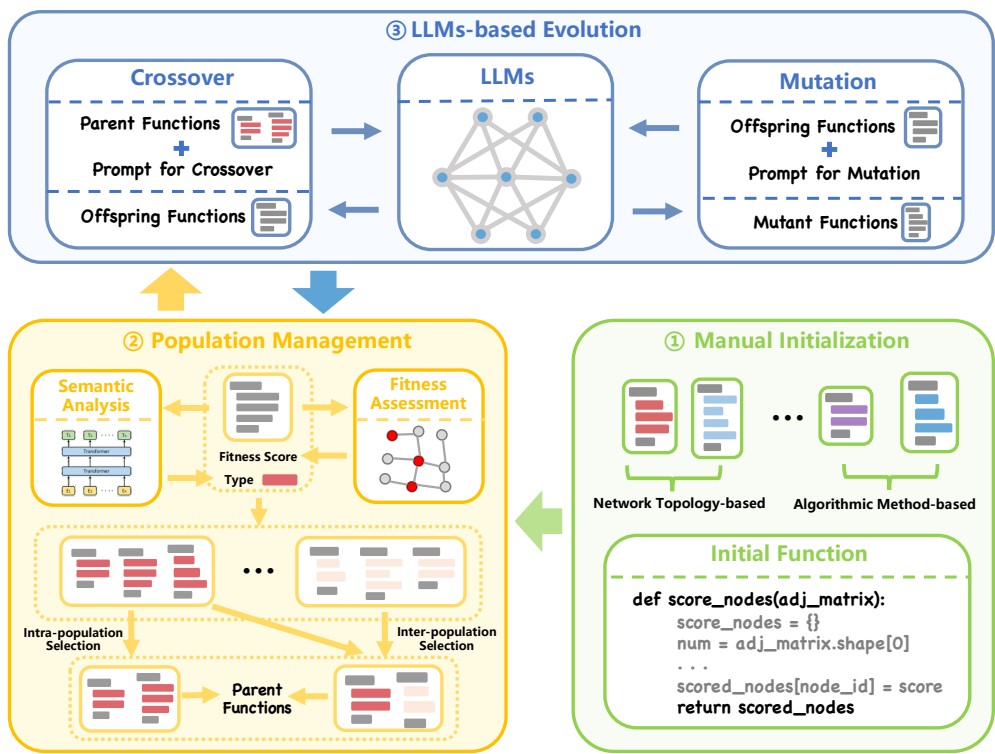

Figure 1: Illustration of our proposed model.

- **Network Topology-based Functions:** Network topology encompasses the connectivity relationships among nodes, and critical nodes often exhibit distinct connectivity patterns compared to other nodes. In this section, we assign corresponding scores to nodes based on their centrality metrics, such as the number of N-hop neighbors of a node, the number of shortest paths in which a node lies, and the reciprocal of the sum of shortest paths from a node to all other nodes. Because of the availability of comprehensive library functions, some centrality metrics can now be directly utilized. To achieve a balance between the scalability and readability of the node scoring function $f$, we not only rely on concise library functions for node scoring but also design some functions that include detailed computation processes.

- **Algorithmic Method-based Functions:** Many classical algorithms analyze network characteristics through mathematical or statistical methods, and they identify critical nodes to dismantle the network based on the analysis. These methods often incorporate greedy algorithms, the efficiency of which is closely related to the network size and the number of selected nodes. In this section, we retain the core ideas of some classic algorithms and manually rewrite them into an efficient node scoring function as our initial approach. It is worth noting that we did not consider learning-based methods, because they typically require significant computational resources, contradicting our goal of having a concise, clear, and efficient node scoring function.

Based on the two approaches described above, we designed $M$ initial functions $\mathcal{F}_0 = \{f_1, f_2, ..., f_M\}$, each with a different emphasis in scoring nodes, all of which can serve as independent populations $\mathcal{P} = \{P_1, P_2, ..., P_M\}$ to start evolution, where $P_i = \{f_i\}$. The input and output of the node scoring function $f$ have been defined in Section 2 as $f(\mathbf{A}) = \{s_{v_1}, s_{v_2}, ..., s_{v_{|\mathcal{V}|}}\}$.

### 3.2 POPULATION MANAGEMENT

To maintain diversity within the function population, avoid falling into local optima, and simultaneously improve the performance of functions, we implement a sophisticated management strategy for the function populations.

### 3.2.1 EVALUATION

Given a node scoring function $f \in \mathcal{F}$, our model first conducts a fitness assessment, which is closely related to our specific task: identifying critical nodes in the network through the learned function. Therefore, we define the node scoring function's fitness evaluation as follows:

$$E(f) = \phi\left(\mathbf{A}, \text{SORT}\left(f(\mathbf{A})\right)[:L]\right), \tag{1}$$

where SORT represents a function that sorts nodes in descending order based on their scores, $[:L]$ indicates selecting the top $L$ elements from the sorted list, and $\phi$ stands for a function that evaluates the network connectivity after removing given nodes. Through the evaluation function above, each executable node scoring function obtains a suitable fitness score $c_f = E(f)$, which assists us in better-performing selection and managing the function populations.

### 3.2.2 CLASSIFICATION

While some functions have low fitness, they can introduce diverse potential evolutionary directions to the population. Conversely, if high-fitness functions already have numerous similar individuals, their widespread presence often indicates a tendency to get trapped in local optima. Considering the above situation, we further classify the node scoring functions $F$ based on their fitness scores.

Inspired by natural language processing tasks, we utilize pre-trained language model Devlin et al. (2018) to process function $f \in \mathcal{F}$ into embeddings $\mathbf{z}_f$, which contain the syntax structure and semantic information of the code. Subsequently, we calculate the cosine semantic similarity $S(\mathbf{z}_f, \mathbf{z}_{P_i})$ between the function and the existing populations as follows:

$$S(\mathbf{z}_f, \mathbf{z}_{P_i}) = \frac{\mathbf{z}_f \cdot \mathbf{z}_{P_i}}{\|\mathbf{z}_f\| \|\mathbf{z}_{P_i}\|}, \tag{2}$$

$$\mathbf{z}_{P_i} = \frac{1}{|P_i|} \sum_{f \in P_i} \mathbf{z}_f, \tag{3}$$

where $\mathbf{z}_{P_i}$ represents the average of function vectors within population $P_i$. At the same time, we establish a similarity threshold $\tau$. Let

$$i^* = \arg\max_i S(\mathbf{z}_f, \mathbf{z}_{P_i}), \tag{4}$$

if $S(z_f, z_{P_i}) > \tau$, $P_{i^*} = P_{i^*} \cup \{f\}$; otherwise, this distinctive function initiates a new population $P_{\text{new}}$. It is worth noting that each population has a limit on the number of individuals. Once a population reaches its maximum capacity, any newly added function must have a higher fitness score to replace the existing lowest-scoring function, thereby maintaining the population's high performance.

### 3.2.3 SELECTION

We implemented two designs for selection of parent functions within the populations, namely, intra-population selection and inter-population selection.

- **Intra-population Selection:** Considering the diversity of functions, our model selects one individual from each population to serve as intra-population parent functions $\mathcal{F}_{\text{intra}}$ in each selection. At the same time, to maintain elite individuals, the highest-fitness function is also added to $\mathcal{F}_{\text{intra}}$. Then, pairs of functions within $\mathcal{F}_{\text{intra}}$ are combined for subsequent crossover operations. Intra-population selection can integrate different types of functions to assist LLMs discover the characteristics of node scoring of given parent functions, thereby generating new offspring functions.

- **Inter-population Selection:** Taking into account the fitness of functions, two functions are chosen from each same population to serve as inter-population parent functions $\mathcal{F}_{\text{inter}}$ in each selection, allowing subsequent crossover operations to discover differences between functions of the same population but with different fitness scores. In other words, inter-population selection aids LLMs in considering which programming steps lead to different fitnesses of two functions from the same population, thereby focusing more on these key steps to enhance function performance.

In both selection methods, the fitness scores of each function serve as selection weights, encouraging the selection of more excellent individuals. For convenience, we use $\mathcal{F}_{\text{parent}}$ to represent $\mathcal{F}_{\text{intra}} \cup \mathcal{F}_{\text{inter}}$.

## 3.3 LLMs-based Evolution

Given the parent functions $\mathcal{F}_{\text{parent}}$, our model begins the evolution process to generate new functions. The traditional crossover and mutation operators used in programming struggles to make high-quality modifications to functions like human programmers do. In contrast, LLMs trained on extensive data including code, documentation, tutorials, and more, can comprehend complex programming tasks and perform high-quality code modifications and generation. Therefore, we use LLMs as the operators to cross and mutate the given functions to generate offspring codes.

### 3.3.1 Crossover

The inputs for crossover consist of a set of selected parent functions $\mathcal{F}_{\text{parent}}$, and the outputs are a set of offspring functions $\mathcal{F}_{\text{off}}$. The prompt provided for LLMs include the functions and their scores, as well as a text-formed description of crossover requirement and additional format-related constraints. We aim for the LLMs to focus on analyzing the distinctions and relationships among functions. Therefore, we have not provided any descriptions related to the network or the task of identifying critical nodes, to avoid causing any interference. The formula for the crossover is shown below:

$$\mathcal{F}_{\text{off}} = \text{LLM}(\mathcal{F}_{\text{parent}}, C_{\text{parent}}, prompt_{\text{crossover}}), \tag{5}$$

where $C_{\text{parent}} = \{c_f, \forall f \in \mathcal{F}_{\text{parent}}\}$ represent the set of fitness scores of all parent functions.

### 3.3.2 Mutation

For the diversity of the populations, each offspring function $f \in \mathcal{F}_{\text{off}}$ has a certain probability $\theta$ of undergoing mutation. The prompts provided as inputs to LLMs include the offspring function and a text-formed description of mutation guiding and format requirements, and the output is a set of mutant functions $\mathcal{F}_{\text{mutate}}$. Similarly, to keep LLMs focused on the code, our prompts do not contain any descriptions related to the task background. The mutation formula is as follows:

$$\mathcal{F}_{\text{mutate}} = \text{LLM}(\mathcal{F}_{\text{off}}, prompt_{\text{mutate}}). \tag{6}$$

Figure 2 illustrates the detailed prompts used in the crossover and mutation. The generated functions $\mathcal{F}_{\text{off}}$ and $\mathcal{F}_{\text{mutate}}$, after undergoing executable checks, are systematically added to the populations following the evaluation and classification described in Section 3.2. This process repeats iteratively, with the entire function population gradually growing until an ideal function emerges.

**Prompt for Crossover**

I have two functions as follows:
Function 1 with < Fitness Score>:

< Parent Function>

Function 2 with < Fitness Score>:
< Parent Function>

Mix the two functions above, and create a Python function called " score_nodes " that accepts " adj_matrix " as input and returns "scored nodes" as output. "adj_matrix" should be a adjacency matrix in the form of a NumPy array, and "scored_nodes" should be a dictionary where the keys are node IDs and the values are node scores.

Provide a Python function only, not any explanation.

**Prompt for Mutation**

Without changing the input and output of this function, be free to adjust the content of the function by symbol, value and logic:

<Offspring Function>

Provide a Python function only, not any explanation.

Figure 2: Prompts for crossover and mutation.

## 4 Experiments and Results

In this section, we aim to comprehensively evaluate the proposed model by providing detailed responses to the following research questions (RQs):

- RQ1: Can our proposed method outperform existing baselines in terms of performance?

- RQ2: Which functions does our model discover?

- RQ3: Through what processes does our model derive these functions?

- RQ4: How effective is each component within our proposed model?

### 4.1 EXPERIMENTAL SETTINGS

#### 4.1.1 EXPERIMENTAL ENVIRONMENT

We compared the performance of our best node scoring functions obtained by our method with four sets of competitive methods on both real-world and synthetic networks. The entire model is implemented in Python and runs on Intel Xeon Platinum 8358. We use the hyperparameters specified in Appendix A.1 for running our method. Each dataset typically requires approximately 2 hours of computation with GPT-3.5-Turbo, costing around 2 dollars per run.

#### 4.1.2 METRICS

Given a network $\mathcal{G} = (\mathcal{V}, \mathcal{E})$, and a predefined connectivity measure $\sigma$, our goal is to identify a sequence of critical nodes $(v_1, v_2, \cdots, v_N)$ to be removed, which can minimize the accumulated normalized connectivity (ANC) defined as follows:

$$R(v_1, v_2, \cdots, v_N) = \frac{1}{N} \sum_{k=1}^{N} \frac{\sigma(\mathcal{G} \backslash \{v_1, v_2, \cdots, v_k\})}{\sigma(\mathcal{G})}, \tag{7}$$

where $N$ represents the number of removed nodes. We consider two most commonly used connectivity measures: (1) the size of giant connected component (gcc) $\sigma_{gcc}(\mathcal{G}) = max\{|C_i|; C_i \in \mathcal{G}\}$, where $C_i$ represents the $i$th connected component in network $\mathcal{G}$, and $|C_i|$ is number of nodes in $Ci$; (2) pairwise connectivity (pc) $\sigma_{pc}(G) = \sum_{C_i \in \mathcal{G}} \frac{|C_i|(|C_i|-1)}{2}$. We utilize both $ANC_{gcc}$ and $ANC_{pc}$ to evaluate the critical nodes obtained by each method.

#### 4.1.3 BASELINES

We compare our proposed method with four sets of baselines, all of which are outlined as follows. (1) Heuristic method: DC. (2) Approximated optimization algorithms: GNDR Ren et al. (2019), Min-Sum Braunstein et al. (2016), CoreHD Zdeborová et al. (2016), VE Huang et al. (2024). (3) Learning-based methods: GDM Grassia et al. (2021), NIRM Zhang and Wang (2022). (4) LLMs-based method: AEL Liu et al. (2023). The introduction of these baselines is in Appendix A.2.

#### 4.1.4 DATASETS

We selected the six different datasets to evaluate our method and baselines, including Jazz Gleiser and Danon (2003), Twitter Fink et al. (2023), Network Science Newman (2006), Synthetic Network, Power Grid Watts and Strogatz (1998), and LastFM Rozemberczki and Sarkar (2020), the statistics of which are detailed in Table 1. Each dataset is detailed in A.3.

Table 1: Statistics of networks.

|  | Jazz | Twitter | Network Science | Synthetic Network | Power Grid | LastFM |
|---|---|---|---|---|---|---|
| $|\mathcal{V}|$ | 198 | 475 | 1,565 | 1,000 | 4,941 | 7,624 |
| $|\mathcal{E}|$ | 2,742 | 13,289 | 13,532 | 2,991 | 6,594 | 27,806 |

### 4.2 OVERALL PERFORMANCE COMPARISON (RQ1)

First, we implement our proposed model and baseline methods on each dataset, selecting 20% of the total nodes from each network. Following this, we remove these nodes from the network and compute the $ANC_{gcc}$ and $ANC_{pc}$ for each case. Table 2 represents a comparison of the overall performance under the $ANC_{gcc}$. Based on these results, we make the following observations.

- **The performance of baselines varies across networks.** The effectiveness of baseline methods differs significantly among networks, where a smaller ANC value reflects a more substantial impact on network connectivity. The method with the best overall performance are GNDR, GDM, and Min-Sum. GNDR ranks second in Jazz, Network Science, Power Grid, and even best in LastFM, but ranks eighth in Sythetic Network. GDM consistently ranks in the top 5 across all datasets but does not have an outstanding performance. Min-Sum, similar to GNDR, performs well on other datasets but has the worst performance in Jazz.

Table 2: Overall performance comparison under the $ANC_{gcc}$.

| Methods | Jazz | Network Science | Twitter | Synthetic Network | Power Grid | LastFM | Average Rank |
|---------|------|-----------------|---------|-------------------|------------|--------|--------------|
| Ours | 0.549±0.008 | 0.025±0.016 | 0.801±0.002 | 0.480±0.005 | 0.012±0.007 | **0.369±0.020** | 1.2 |
| DC | 0.887 | 0.155 | 0.899 | 0.811 | 0.307 | 0.722 | 7.5 |
| GNDR | **0.827** | **0.055** | 0.897 | 0.833 | **0.052** | 0.335 | **3.0** |
| Min-Sum | 0.890 | 0.084 | 0.898 | **0.786** | 0.086 | 0.596 | 4.3 |
| CoreHD | 0.886 | 0.153 | 0.899 | 0.853 | 0.462 | 0.688 | 7.3 |
| WN | 0.886 | 0.163 | 0.899 | 0.853 | 0.406 | 0.700 | 7.5 |
| VE | 0.862 | 0.133 | 0.899 | 0.816 | 0.279 | NaN | 6.3 |
| GDM | 0.884 | 0.064 | 0.899 | 0.796 | 0.222 | 0.676 | 4.0 |
| NIRM | 0.886 | 0.093 | 0.899 | 0.809 | 0.269 | 0.708 | 5.8 |
| AEL | 0.882 | 0.094 | **0.896** | 0.798 | 0.300 | 0.757 | 5.3 |

Data highlighted in  yellow  denotes the method that performs best in the network, while the **bolded** data's method performs second best. Number in ( ) is the ranking of the data, and a smaller ANC indicates better performance.

- **Superior Overall Performance of Our Model.**

  Our model distinguishes itself by surpassing all competitors in Jazz, Network Science, Twitter, Synthetic Network, and Power Grid by a notable margin, with only a slight difference from GNDR, still significantly ahead of other methods. The uniform excellence of our model across diverse networks suggests its capability to more accurately score nodes, thereby more effectively identifying critical nodes within any given network.

The detailed results under the $ANC_{pc}$ are shown in Appendix A.4, and our method still holds the highest overall ranking.

### 4.3 FUNCTION ANALYSIS (RQ2)

We present the best node scoring functions derived from the Jazz and Network Science datasets in Figure 3. For Jazz, the function assesses nodes by integrating local features with centrality measures. It begins by computing the network's largest connected component to derive a Laplacian matrix (lines 4, 9–10). Utilizing this matrix, the function proceeds to compute the Fiedler vector, employing it to establish a new graph (lines 11–13). This new graph identifies a minimum weighted cover to assign scores to its nodes (lines 13–22). Nodes not included in this new graph receive scores based on normalized eigenvector centrality (lines 5–8, 24).

Conversely, the best function from the Network Science dataset determines node scores by calculating degree-related metrics (lines 4–8), betweenness centrality (line 9), and PageRank values (line 10) independently. These metrics are then aggregated using weighted averages to calculate a comprehensive node score (lines 11–19), effectively merging various centrality measures for node evaluation.

This analysis reveals that the scoring functions discovered by our model differ across networks, all originating from initial functions and undergoing evolution and diverse modifications to logically score nodes.

### 4.4 CASE STUDY (RQ3)

Figure 4 illustrates the statistics of function populations during the training process in Synthetic Network, aiding us in understanding the evolutionary process of the function populations. Figure **(a)**, **(b)** and **(c)** respectively represent the evolution of the population size, the average fitness score, and the highest fitness score of each population.

First, in terms of population size, as evolution progresses, the majority of populations gradually increase, reaching capacities and updating via elimination. However, for Population 2, the function size consistently remains at one, indicating the absence of offspring function generation. Given that Population 2's fitness score is the lowest among all initial functions, it logically follows that its *inferior genes* are not selected by our model. Furthermore, two new populations emerged during evolution, illustrating that our population management design can preserve the diversity among function populations.

```
Best function in Jazz
1  def score_nodes(adj_matrix):
2      adj_matrix = np.array(adj_matrix)
3      G = nx.from_numpy_array(adj_matrix)
4      LCC = G.subgraph(max(nx.connected_components(G), key=len))
5      eigenvalues, eigenvectors = np.linalg.eig(adj_matrix)
6      max_eigenvalue_index = np.argmax(eigenvalues)
7      eigenvector = eigenvectors[:, max_eigenvalue_index]
8      normalized_eigenvector = eigenvector / np.sum(eigenvector)
9      ii = {v: i for i, v in enumerate(LCC.nodes())}
10     L = nx.normalized_laplacian_matrix(LCC)
11     eigenvalues, eigenvectors = eigsh(L, which='SM', maxiter=1000 * L.shape[0])
12     Fiedler = eigenvectors[:, 1]
13     H = nx.Graph([(u, v) for u, v in LCC.edges() if Fiedler[ii[u]] * Fiedler[ii[v]] <= 0.0])
14     for v in H.nodes():
15         H.nodes[v]['weight'] = 1.0 / H.degree(v)
16     cover = list(nx.algorithms.approximation.min_weighted_vertex_cover(H, weight='weight'))
17     max_degree = max([G.degree(v) for v in G.nodes() if v not in cover])
18     min_weight = min(H.nodes[v]['weight'] for v in H.nodes())
19     scored_nodes = {}
20     for v in G.nodes():
21         if v in cover:
22             scored_nodes[v] = H.nodes[v]['weight']
23         else:
24             scored_nodes[v] = normalized_eigenvector[v] * min_weight
25     return scored_nodes
```

```
Best function in Network Science
1  def score_nodes(adj_matrix):
2      G = nx.Graph(adj_matrix)
3      score = {node_id: 0.0 for node_id in G.nodes()}
4      for u, v in G.edges():
5          degree_u = G.degree(u)
6          degree_v = G.degree(v)
7          score[u] += 1.0 / degree_u if degree_u != 0 else 0.0
8          score[v] += 1.0 / degree_v if degree_v != 0 else 0.0
9      betweenness = nx.betweenness_centrality(G)
10     pagerank = nx.pagerank(G)
11     weight_degree = 0.2
12     weight_betweenness = 0.3
13     weight_pagerank = 0.5
14     scored_nodes = {}
15     for node_id in G.nodes():
16         score = (weight_degree * score.get(node_id, 0) +
17             weight_betweenness * betweenness[node_id] +
18             weight_pagerank * pagerank[node_id])
19         scored_nodes[node_id] = score
20     return scored_nodes
```

Figure 3: The best functions found in Jazz and Network Science.

Moreover, the analysis of changes for average and highest fitness scores across populations reveals continuous enhancement in the overall function quality. Simultaneously, no instances of multiple populations maintaining the same average fitness throughout evolution were observed, indicating our design's effectiveness in circumventing the pitfall of convergent evolution. Around the 20th and 100th epochs, there are noticeable increases in the fitness scores of the function populations, representing the circulation of *superior genes* within the populations and enhancing overall performance.

In conclusion, our proposed model ensures that the function populations can evolve continuously while preserving diversity, thereby facilitating the generation of superior new functions.

## 4.5 ABLATION STUDY (RQ4)

Our model is built upon three key components, and we conduct ablation studies to evaluate the effectiveness of each component, including (1) W/O Manual Initialization, (2) W/O Population Management, and (3) W/O LLMs-based Evolution. The specific setup of the ablation experiment and additional experimental results are provided in Appendix A.5. Here, we present the ablation experiment results under the $\text{ANC}_{gcc}$ metric in Table 3.

By comparing the performance of three ablation models with the complete model across different datasets, we found that the complete model significantly outperforms the others. This affirms that each component of our proposed model is essential and significantly enhances overall performance.

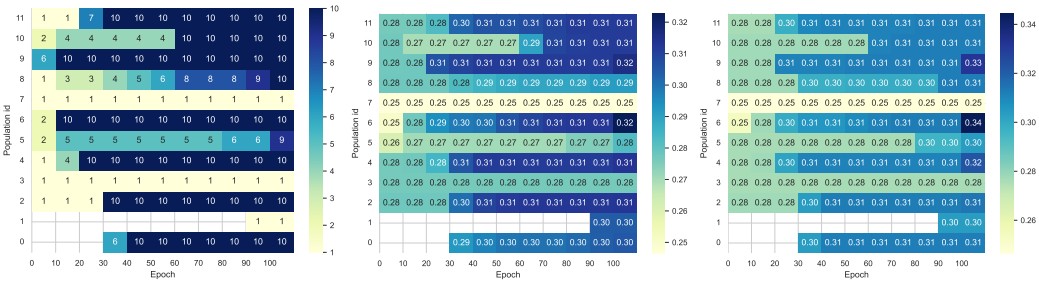

(a) Population Size Heatmap  (b) Average Fitness Score Heatmap  (c) Highest Fitness Score Heatmap

Figure 4: Evolution of funciton populations.

Table 3: Comparison of ablation study under the $ANC_{gcc}$.

| Model | Jazz | Network Science | Twitter | Synthetic Network | Power Grid | LastFM |
|---|---|---|---|---|---|---|
| Complete Model | 0.549 | 0.025 | 0.801 | 0.480 | 0.012 | 0.369 |
| W/O Manual Initialization | 0.888 | 0.125 | 0.899 | 0.821 | 0.290 | 0.745 |
| W/O Population Management | 0.874 | 0.093 | 0.898 | 0.803 | 0.125 | 0.620 |
| W/O LLMs-based Evolution | 0.886 | 0.149 | 0.899 | 0.843 | 0.257 | 0.694 |

## 5 RELATED WORKS

### 5.1 SYNERGY BETWEEN LLMs AND EAs

Recent research underscores the effectiveness of leveraging LLMs for optimization tasks via prompt engineering Yang et al. (2023); Guo et al. (2023a). This method proves particularly advantageous for EAs, attributed to the advanced text comprehension and generation capabilities of LLMs Chao et al. (2024); Wu et al. (2024). For example, recent investigations into discrete prompt optimization have shown how LLMs can mimic evolutionary operators, generating innovative candidate prompts through crossover and mutation based on fixed prompts Guo et al. (2023b); Li and Wu (2023). Furthermore, studies have explored the use of LLMs in tackling combinatorial optimization problems (COPs) by developing evolved heuristic solutions Pluhacek et al. (2023); Liu et al. (2023; 2024); Ye et al. (2024). EvoPrompting Chen et al. (2023) utilizes LLMs for Neural Architecture Search (NAS), performing crossover and mutation at the code level to discover new architectures. Likewise, FunSearch Romera-Paredes et al. (2023) implements an LLM-powered evolutionary method to determine solution functions for mathematical challenges. In addition, Eureka Ma et al. (2023) focuses on the design of reinforcement learning reward functions using LLM-based evolutionary algorithms, while Fuzz4All Xia et al. (2024) applies LLMs in fuzz testing to generate and mutate inputs.

Distinct from previous studies, our research emphasizes the use of LLMs that perform mutation and crossover processes on node scoring functions, aiming to identify critical nodes in complex networks.

### 5.2 CRITICAL NODES IDENTIFICATION

Identifying critical nodes is of significant importance across various practical domains. Basic methods such as degree centrality entail removing nodes with the highest degrees. Algorithms like CoreHD Zdeborová et al. (2016) and WeakNeighbors Schmidt et al. (2019) utilize $k$-core decomposition, in which the $k$-core of a network is defined as the maximal subgraph with every node having a degree of at least k. These methods progressively eliminate nodes with the highest degrees within their respective $k$-cores to identify key nodes. On the other hand, Min-Sum Braunstein et al. (2016) focuses on creating an acyclic network by removing nodes that form loops, followed by utilizing greedy tree-breaking algorithms to decompose the remaining forest into smaller disconnected components. Additionally, GND Ren et al. (2019) adopts a spectral approach to iteratively partition the network into two components. Learning-based methods like NIRM Zhang and Wang (2022) and GDM Grassia et al. (2021) utilize neural networks to assign scores to network nodes, determining their importance based on these scores.

Our research is not limited to identifying specific algorithms. Rather, it leverages LLMs-empowered EAs to investigate a broad spectrum of potential algorithms in all kinds of networks.

## 6 CONCLUSIONS AND LIMITATIONS

In this study, we employ an evolutionary algorithm enhanced by LLMs for generating node scoring functions, to identify critical nodes within networks. Experimental results across various network datasets reveal that our method effectively balances adaptability and utility in comparison to other state-of-the-art algorithms. Moreover, it is capable of generating a variety of both reasonable and efficient node scoring functions for identifying critical nodes.

Our current model undergoes sequential training, which is relatively time-consuming. In future work, we aim to delve deeper into distributed designs for the evaluation, management, and evolution processes, enabling parallel execution to improve the model's evolutionary speed and performance.

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

# A APPENDIX

## A.1 EXPERIMENTAL SETTINGS

Table 4 shows the hyperparameter settings used in the experiment.

Table 4: Hyperparameters of proposed method.

| Hyperparameter | Value |
|---|---|
| LLMs (crossover and mutation) | gpt-3.5-turbo-0613 |
| Code2Vec model | bert-base-uncased |
| Number of initial populations | 10 |
| Maximum population size | 10 |
| Maximum number of epoch | 100 |
| Mutation rate | 0.3 |
| Similarity threshold | 0.93 |

The similarity threshold $\tau$ is used for generated function classification. If $\tau$ is set too high, it becomes difficult for generated functions to find a classification and then new populations are created, leading to an increase in the number of populations and impacting evolutionary efficiency. On the other hand, if $\tau$ is set too low, it is challenging to create brand new populations, which is not conducive to maintaining population diversity. We conducted a statistical analysis of the number of final populations at different $\tau$ in the Twitter dataset, the results of which are shown in Table 5. It is observed that as $\tau$ increases, number of final populations gradually increases. To maintain diversity and prevent population overgrowth, we set $\tau$ to 0.93.

Table 5: Number of final populations at different similarity threshold.

| Similarity threshold | 0.7 | 0.8 | 0.9 | 0.91 | 0.92 | 0.93 | 0.94 | 0.95 | 0.96 |
|---|---|---|---|---|---|---|---|---|---|
| Number of final populations | 10 | 10 | 10 | 10 | 10 | 11 | 13 | 16 | 22 |

## A.2 DETAILED INFORMATION ABOUT BASELINES

- **DC (Degree Centrality)**: A heuristic method that iteratively removes nodes with the highest degrees.

- **GNDR** Ren et al. (2019): GNDR employs a spectral approach to iteratively partition the network into two components.

- **Min-Sum** Braunstein et al. (2016): It initially creates a cyclic network by removing nodes that form loops, then employs greedy tree-breaking algorithms to decompose the resultant forest into smaller, disconnected components.

- **CoreHD** Zdeborová et al. (2016): CoreHD utilizes a $k$-core decomposition strategy, where the $k$-core of a network denotes the maximal subgraph in which every node has a degree of at least k. CoreHD progressively removes nodes with the highest degrees within its $k$-core to select key nodes.

- **WN (Weak Neighbors)** Schmidt et al. (2019): WN improves upon CoreHD by selecting, for deletion, the neighbor node with the minimum degree when multiple nodes have the maximum degree in a $k$-core. Conversely, CoreHD randomly selects one of the nodes with the maximum degree within a $k$-core for deletion.

- **VE (Vertex Entanglement)** Huang et al. (2024): VE incorporates concepts from quantum mechanics, utilizes an entanglement-based metric to quantify the perturbations that individual vertices induce in spectral entropy, and identifies key nodes based on these measurements.

- **GDM (Graph Dismantling with Machine learning)** Grassia et al. (2021): A machine learning-based approach that uses graph convolutional networks to understand the topological structure of networks and to evaluate the likelihood of node attacks.

- **NIRM (Neural Influence Ranking Model)** Zhang and Wang (2022): A machine learning-based method conducting a comprehensive assessment of both global and local node features in the network, effectively identifying critical nodes.
- **AEL** Liu et al. (2023): An approach called Algorithm Evolution with Large Language Model (AEL) for automatic algorithm design. We adhere to the original paper to reproduce AEL.

## A.3 DETAILED INFORMATION ABOUT DATASETS

- **Jazz** Gleiser and Danon (2003): A collaboration network among Jazz musicians, where each node corresponds to a musician, and an edge signifies that two musicians have collaborated in a band.
- **Twitter** Fink et al. (2023): A dataset quantifies the pairwise probability of influence within a Congressional Twitter network.
- **Network Science** Newman (2006): A co-authorship network within the field of network science.
- **Synthetic Network**: A Barabási-Albert ($m = 3$) network, randomly generated with 1000 nodes.
- **Power Grid** Watts and Strogatz (1998): An undirected network containing information about the power grid of the USA.
- **LastFM** Rozemberczki and Sarkar (2020): A social network of LastFM users from Asian countries.

## A.4 ADDITIONAL EXPERIMENTAL RESULTS OF RQ1

Table 6 represents a comparison of the overall performance under the $\text{ANC}_{pc}$. Based on these results, we still make the conclusions described in 4.2: the performance of baselines varies across networks and our model holds the highest overall ranking.

Table 6: Overall Performance Comparison under the $\text{ANC}_{pc}$.

| Method | Jazz | Network Science | Twitter | Synthetic Network | Power Grid | LastFM | Average Rank |
|--------|------|-----------------|---------|-------------------|------------|--------|--------------|
| Ours | 0.663±0.035 | **0.053±0.024** | **0.806±0.002** | **0.655±0.011** | 0.113±0.008 | 0.492±0.005 | 1.8 |
| DC | 0.791 | 0.118 | 0.811 | 0.677 | 0.203 | 0.546 | 7.7 |
| GNDR | **0.697** | **0.053** | 0.808 | 0.706 | **0.142** | 0.262 | **3.0** |
| Min-Sum | 0.794 | 0.052 | 0.805 | 0.656 | 0.153 | **0.447** | 3.5 |
| CoreHD | 0.788 | 0.099 | 0.811 | 0.736 | 0.350 | 0.530 | 7.3 |
| WN | 0.788 | 0.103 | 0.811 | 0.738 | 0.321 | 0.534 | 7.7 |
| VE | 0.748 | 0.086 | 0.811 | 0.683 | 0.147 | NaN | 5.8 |
| GDM | 0.784 | 0.059 | 0.811 | 0.654 | 0.165 | 0.493 | 4.2 |
| NIRM | 0.789 | 0.068 | 0.811 | 0.675 | 0.187 | 0.529 | 5.8 |
| AEL | 0.782 | 0.054 | 0.809 | 0.657 | 0.190 | 0.587 | 5.3 |

Data highlighted with yellow means its method performs the best on the network, while the **bolded** data's method is the second best. Number in ( ) is the ranking of the data, and smaller ANC indicates better performance.

## A.5 EXPERIMENTAL DETAILS AND ADDITIONAL EXPERIMENTAL RESULTS OF RQ4

Table 7: Comparison of Ablation Study under the $\text{ANC}_{pc}$.

| Model | Jazz | Network Science | Twitter | Synthetic Network | Power Grid | LastFM |
|-------|------|-----------------|---------|-------------------|------------|--------|
| Complete Model | 0.663 | 0.053 | 0.806 | 0.655 | 0.113 | 0.492 |
| W/O Manual Initialization | 0.768 | 0.064 | 0.811 | 0.666 | 0.145 | 0.501 |
| W/O Population Management | 0.724 | 0.078 | 0.811 | 0.692 | 0.153 | 0.524 |
| W/O LLMs-based Evolution | 0.789 | 0.089 | 0.811 | 0.683 | 0.178 | 0.534 |

We designed the ablation models according to the following methods:

- **W/O Manual Initialization:** To gauge the significance of Manual Initialization, we adopt an alternate strategy. Here, we define the task of identifying critical nodes to the LLM and instruct

it to generate an equivalent number of node scoring functions in a predetermined format, thereby bypassing manual initialization.

- **W/O Population Management:** To explore the necessity of Population Management, we implement an alternative scheme where all functions are collected without any categorization. During the selection phase, parent functions are randomly selected from the entire pool of individuals, eliminating structured population management.

- **W/O LLMs-based Evolution:** Assessing the necessity of LLMs-based Evolution, we introduce an alternative method. Initially, all functions are treated as parent functions. This is followed by a single epoch of crossover and mutation to produce a multitude of offspring functions, from which the most suitable are chosen, simulating a condensed version of the evolutionary process.

Here, we still remove 20% of the nodes in each network based on node scoring function obtained by ablation models, and then calculate $ANC_{gcc}$ and $ANC_{pc}$. The ablation results under the $ANC_{pc}$ metric in Table 7.

