# OpenReview forum: "Identify Critical Nodes in Complex Network with Large Language Models"
_ICLR.cc/2025/Conference — Submitted to ICLR 2025_

### Official Review · Reviewer_pCWG · 2024-10-28

**Soundness:** 2
**Presentation:** 3
**Contribution:** 2
**Rating:** 3
**Confidence:** 5

**Summary:**

This work use LLMs and genetic algorithm to generate the algorithms for the identification influential nodes in complex networks and the results shows the LLMs-generated algorithm is better than some of existing approaches across different datasets.

**Strengths:**

1. This work leverages the NLP techniques to facilitate the LLMs-based algorithm generation, which may provide some insights to the related community.

2. The algorithm found by this work may provide some insights when designing efficient heuristic in solving network science tasks.

**Weaknesses:**

1. The primary concern is the generalizability of this approach. While pioneering studies like [1,2] have demonstrated effectiveness across various combinatorial problems, this work is limited to a single-task focus. With this in mind, I suggest expanding the evaluation to include additional tasks, particularly in the area of complex networks, such as influence maximization and immunization.

2. The contribution is limited as there are already several works of LLMs-based algorithm generation and it would be good to see if there are more contributions in terms of structural design instead of just an application on a different task. 3.2.1 is a normal metric in network science area and 3.2.3 is very similar to [2].

3. The tested problem is also known as graph dismantling where the curve of the largest connected component should be plotted as the part of experiment.

4. Some key experimental setting is missing, such as crossover rate.

5. The citation format should be improved.

[1] Liu, Fei, et al. "Evolution of Heuristics: Towards Efficient Automatic Algorithm Design Using Large Language Model." Forty-first International Conference on Machine Learning. 2024.

[2] Romera-Paredes, Bernardino, et al. "Mathematical discoveries from program search with large language models." Nature 625.7995 (2024): 468-475.

**Questions:**

As both LLMs and evolutionary algorithms are with randomness, I am quite curious about the stability of the searched algorithm. Is it always able to find the algorithm achieving the same result as reported? Is there any statistic analysis?

---

### Official Review · Reviewer_6u9d · 2024-11-03

**Soundness:** 2
**Presentation:** 2
**Contribution:** 2
**Rating:** 5
**Confidence:** 3

**Summary:**

This paper introduces a new mechanism for generating scoring functions to identify critical nodes in networks. Specifically, the authors employ EA and LLM to produce node scoring functions in an optimization manner. The approach begins with an enhanced population initialization strategy. Following this, an LLM-based variation mechanism is implemented to evolve scoring functions. Finally, an inter- and intra-population selection strategy is applied to select the most promising functions for the next iteration. Experimental results demonstrate the superior performance of the proposed method.

**Strengths:**

1. The authors utilize Evolutionary Algorithms (EA), a gradient-free optimization method, for this domain-specific application.
2. The authors introduce a novel approach to the variation step in Evolutionary Algorithms (EA) by leveraging Large Language Models (LLMs), extending their use beyond traditional question-answering applications.
3. Figures 1 and 2 are well designed to help readers understand the workflow of the proposed method.
4. Research Questions 1–4, outlined in Section 4, clearly show readers the structure of the subsequent content.

**Weaknesses:**

1. This paper integrates several existing methods, including LLMs and EAs. While the authors adapt EAs specifically for their critical node detection problem, the challenges and difficulty of this customization are not thoroughly demonstrated.
2. The expressions in the paper could benefit from more precise, domain-specific terminology.
For example, to the best of my knowledge, ‘combinatorial optimization’ and ‘evolutionary learning’ are more commonly accepted terms than ‘combinational optimization’ and ‘evolutional learning’. Phrases like ‘optimal search task’ are somewhat unclear. Using well-established terms from optimization theory or operations research would make the text more readable and aligned with standard terminology in the field.
3. Several established methods from the current field of Evolutionary Algorithms (EA) could be leveraged in this paper. For instance, the subset selection problem is a well-known combinatorial optimization task, and related optimization algorithms might be applicable to this paper’s objectives. However, the discussion of existing EA research relevant to this optimization task is missing in the current paper.
4. The authors employ a multi-population strategy during optimization, which is uncommon in Evolutionary Algorithms (EA). Providing a discussion on the motivation behind this choice, along with evidence of its benefits, would strengthen the paper and enhance the reader's understanding.
5. Pseudocode or a diagram regarding the overall workflow of this work's method is not provided.

**Questions:**

1. The writing could be refined with more precise and widely accepted terms from the literature.
2. In line 106 of page 2, the authors state that a well-designed population initialization strategy can help identify optimal solutions, particularly for large-scale tasks with a limited population size. However, it is important to note that in many cases, initialization strategies primarily enhance convergence speed rather than directly improving solution quality. It would be beneficial for the authors to provide citations supporting this claim. Additionally, it is unclear whether the proposed problem is indeed large-scale.
3. Several technical aspects require further clarification, including the motivation for using multiple populations and the challenges associated with applying existing EAs to the proposed problem. Additionally, the LLMs may generate inappropriate or even erroneous prompts for crossover and mutation operations. It is unclear how the authors address these potential issues in their approach.
4. Including a workflow diagram/pseudocode.

**Details Of Ethics Concerns:**

I do not identify any ethical concerns

---

### Official Review · Reviewer_3KLm · 2024-11-03

**Soundness:** 2
**Presentation:** 1
**Contribution:** 2
**Rating:** 3
**Confidence:** 3

**Summary:**

This paper presents an approach combining Large Language Models (LLMs) with evolutionary algorithms to generate scoring functions for identifying critical nodes in networks. The paper proposes:

- A framework that uses LLMs to generate and evolve functions that assign importance scores to network nodes
- A population management system that classifies functions using semantic similarity and implements both intra- and inter-population selection
- An evolution mechanism where LLMs perform crossover and mutation operations on the scoring functions
- Empirical results comparing the method against 9 baseline approaches across 6 networks using two metrics (ANCgcc and ANCpc)
- Ablation experiments examining the effects of removing each component (manual initialization, population management, LLM-based evolution)

The work includes performance benchmarks on several network datasets (Jazz, Twitter, Network Science, Synthetic Network, Power Grid, LastFM), analysis of the generated scoring functions, and documentation of the implementation including hyperparameters and experimental configuration. The authors aim to address what they term the "adaptability-utility dilemma" in critical node identification, where methods typically trade off between adaptability to different network types and computational efficiency.

**Strengths:**

1. The paper presents an innovative reformulation of critical node identification as a code generation task guided by LLMs and evolutionary algorithms, demonstrating a novel approach to this NP-hard problem
2. The empirical results across diverse network types (from social networks to power grids) suggest this paradigm of using LLMs to generate and evolve scoring functions could be valuable for similar combinatorial optimization problems
3. By combining LLMs with evolutionary search, the paper introduces an interesting framework for exploring the space of possible node scoring functions

**Weaknesses:**

1. The method's practical value is undermined by inadequate runtime analysis. While the approach achieves good accuracy metrics, GNDR achieves strong performance (average rank 3.0) across datasets with much lower computational requirements. Without any analysis of end-to-end runtime comparisons, it's difficult to evaluate whether the additional complexity is warranted.

2. The ablation study comparing against a single epoch of traditional evolutionary operators provides insufficient evidence for the value of LLMs in this context. While the constraints of traditional evolutionary operators may differ from LLM-based ones, a more thorough comparison should include multiple epochs of traditional evolution (if computationally feasible) and analysis of how LLM-based operations differ qualitatively from traditional genetic operators.

3. Several key methodological components are inadequately described. The criteria for splitting initial functions between topology-based and algorithm-based approaches is unspecified, the population management mechanism's specific implementation is omitted beyond high-level descriptions, and the exact prompting strategy and constraints for LLM-based evolution are not detailed. Perhaps all the necessary details are included, but a full algorithm for the entire procedure is not present anywhere I could see.

4. Despite presenting a related work section on LLM-guided evolution and optimization (e.g., Guo et al., Li and Wu), the authors don't compare against or build upon these approaches. The paper discusses numerous relevant techniques for LLM-based evolutionary operators and combinatorial optimization but doesn't justify why their specific approach is preferable to or different from these established methods. This makes it difficult to assess the true contribution beyond applying existing techniques to a new domain.

**Questions:**

1. Could you provide a runtime analysis comparing your method against baseline approaches, particularly GNDR? Given the additional complexity from LLM operations, evolutionary epochs, and population management, understanding the computational tradeoffs relative to a simpler spectral method seems crucial for evaluating practical utility.
2. Could you justify comparing against only a single epoch of traditional evolution in the ablation study? A more thorough comparison showing how LLM-based operations differ qualitatively from traditional genetic operators over multiple epochs seems necessary to demonstrate the value of LLMs in this context.
3. Could you provide the missing implementation details for the (a) criteria for splitting initial functions between topology-based and algorithm-based approaches, (b) clarification regarding how your crossover+mutation population management strategy differs from those outlined in your related work section, and (c) exact prompting strategy and constraints used for LLM-based evolution, for example whether the resulting LLM output was always a valid function, and what you did in that case?
4. More generally, given the related work you discuss on LLM-guided evolution and optimization (e.g., Guo et al., Li and Wu), how does the EA part of your approach compare to or build upon these methods?

---

### Official Review · Reviewer_bkFS · 2024-11-03

**Soundness:** 3
**Presentation:** 1
**Contribution:** 2
**Rating:** 3
**Confidence:** 4

**Summary:**

This paper focuses on using Large Language Models in an Evolutionary Algorithm to write code that can identify critical nodes in complex networks. This paper presents an algorithm that uses LLMs as the mutation and reproduction operators for code in an evolutionary framework. To do this, the authors prompt a language model to combine or mutate given models for use in an evolutionary algorithm. The authors compare the proposed approach to a variety of baselines, and use six different datasets to evaluate the method. The results show relatively consistent improvements for using the proposed approach over prior work.

**Strengths:**

- The paper is relatively easy to follow.
- Comparisons are made to a large number of baselines, proving experimental rigor.
- The results are consistent, and prove the key conclusions of the paper well.
- I appreciate the function analysis and case study, showing that techniques based on code generation can improve on the interpretability of our machine learning methods.

**Weaknesses:**

- While the use-case is novel to my knowledge, the use of an LLM as the mutation operator for an evolutionary algorithm framework is not novel, and has not been adequately framed in relation to prior work (ELM, etc). [1]
- This work seems like a very narrow use case. I would have liked to see whether this method can adapt to other network based domains.
- Although there is an ablation study, I don’t find it convincing as evidence for the use of such as sophisticated population management framework. The authors did not well advocate for their choices in the use of initialization and population management. From the appendix, it appears that the ablation for population management simply removes any selection pressure through the use of random selection. I suggest modifying this ablation to simply use a normal evolutionary algorithm with a population of candidates that is selected from and mutated/reproduced over time.

[1] Lehman, J., Gordon, J., Jain, S., Ndousse, K., Yeh, C. and Stanley, K.O., 2023. Evolution through large models. In Handbook of Evolutionary Machine Learning (pp. 331-366). Singapore: Springer Nature Singapore


Small notes:

- Citations are incorrectly formatted throughout. (citep instead of cite)

- Many small grammatical errors and awkward phrasings
    - e.g “The first kind of work is the basic methods based on heuristics”
    - e.g. “LLMs-based” should be “LLM-based”

- There is a lack of relation to prior work on using evolutionary algorithms for code generation. It would have been nice to comment on the differences between this approach and previous techniques such as Genetic Programming.
- Section 3.2 is difficult to follow.
- The intermixing of singular and plural for population is confusing. For example, in section 3.2 -- is there one population or many throughout evolution? I would recommend changing the name of population (meaning the inner population) to something more fitting like “species”, for example.
- I assume from the phrase “the fitness scores of each function serve as selection weights,” that the authors are using fitness proportionate selection. I would have liked this to be more clear.

**Questions:**

- What does 3.2.2 have to do with "classification”? Isn’t it just a diversity focused fitness augmentation? I would appreciate this being explained in a bit more depth.
- How general can you expect this method to perform? It seems as though you are catering a specific function for each dataset, which strikes me as somewhat brittle. Do you expect that functions found for one task can generalize to others? If so, I would like to see that as an additional result. If not, I would like that to be made more clear.
- Is there any chance that the solutions for the datasets used exist in the LLM training data?

---

### Official Review · Reviewer_kUgr · 2024-11-04

**Soundness:** 3
**Presentation:** 3
**Contribution:** 3
**Rating:** 5
**Confidence:** 4

**Summary:**

This paper proposes a method that combines large language models with evolutionary algorithm concepts to generate a "score_nodes" function. This function can provide evaluation scores for each node and is used to identify critical nodes in complex networks. The method comprises three parts: Manual Initialization, Population Management, and LLMs-based Evolution. Manual Initialization initializes various existing functions. Population Management involves dividing the population based on similarity and evaluating the functions' fitness. Based on the population and fitness, it generates intra-population selection and inter-population selection for each iteration. Large language models are used to analyze similarities and differences within and between groups to generate a new generation of functions.

**Strengths:**

1. Introduces a new perspective by integrating LLMs with EAs to transform the critical node detection problem into a code-generation task.
2. Proposes a general framework that combines LLMs and evolutionary learning, along with a generation-update mechanism for LLMs and crossover-mutation methods for evolutionary learning.

**Weaknesses:**

1. The experimental results show that the author's method significantly outperforms other baselines, while the performance of all other baselines is similar. This raises concerns about the experimental results, as the author does not provide a detailed analysis or explanation of the source of such a significant improvement (i.e., which computational processes contribute to it).

2. Ablation studies indicate that manual initialization has a substantial impact on the method's effectiveness. This raises the question of whether the entire generation process relies on very strong prior knowledge, possibly even containing initialization functions that surpass all baseline methods.

3. The manual initialization process might require significant expertise and could be a bottleneck in applying the model to different networks without prior knowledge.

**Questions:**

I would like to know what the initialization functions include, whether they include a baseline, and if there are initialization functions that exceed baseline performance. I hope to see experimental results that illustrate the ANC_gcc performance of these initialization functions before any iterations.

I would like the author to conduct an ablation study on the best function to analyze which computational process contributes to the significant performance improvement and which initialization function this process is derived from.

---

### Official Review · Reviewer_c56j · 2024-11-04

**Soundness:** 3
**Presentation:** 3
**Contribution:** 3
**Rating:** 6
**Confidence:** 3

**Summary:**

This paper presents a framework combining Evolutionary Algorithms (EAs) and Large Language Models (LLMs) to identify critical nodes in complex networks. The EA’s search space consists of functions that score nodes based on information from the network's adjacency matrix. The framework includes three modules: Manual Initialization, Population Management, and LLM-based Evolution. In the first module, initial scoring functions are generated using network topology or mathematical methods to target nodes whose removal would dismantle the network. The second module assesses functions based on fitness and semantic analysis to select parent functions for the next generation. Finally, the third module uses LLMs for evolution, applying crossover and mutation to generate new functions from the parent functions. Experimental results on real-world and synthetic networks suggest this approach outperforms other baseline methods, producing diverse and effective node-scoring functions.

**Strengths:**

1. The paper presents a novel approach to identifying critical nodes in a network using LLMs for code generation.
2. The authors cast the critical node generation problem as a code generation problem where LLMs can generate node scoring functions that can potentially adapt to different network structures.
3. The authors introduce a population management approach that uses semantic analysis and metrics on function embeddings to maintain diversity in the function population.
4. The proposed approach performs well in the experiments

**Weaknesses:**

1. The authors may consider testing LLM’s robustness for varying network structures and properties and share the insights and the limitations.
2. The authors may consider implementing an interpretability mechanism, such as feature importance analysis, to understand which generated functions are more effective on which network structures.
3. The authors are encouraged to provide a more comprehensive discussion of the limitations of this approach.

**Questions:**

1. How are the prompts developed? How does it impact the generated functions?
2. Are there specific network properties where this approach underperforms compared to baselines, and how could these be addressed?
3. Can this approach be optimized for efficiency so that could make the approach more practical for large-scale applications?

---

### Meta-Review · Area_Chair_qBez · 2024-12-21

**Metareview:**

This paper proposed a framework combining LLMs with Evolutionary Algorithms (EA) to generate  "score_nodes" function to identify the critical nodes in complex networks.

Strengths:
1.  The idea of Integrating LLMs and EAs to address the critical node identification is interesting.

2.  The authors provided results on different network types and metrics, and it shows some potential of the proposed method.

Weaknesses:

1. There is some overlaps with prior LLM-based combinatorial optimization methods.

2. There is a lack of computational complexity analysis.

3. Some concerns were raised on generalization and robustness of the method since current performance improvements seems to rely heavily on manual initialization. Additionally, the ablation studies are insufficient to validate the claimed contributions.

4. The paper is not well-written and some parts are unclear. For example, the population management and prompting strategies are not well explained.

While the idea is interesting, I have to recommend rejection in its current form but I sincerely encourage the authors to revise it before future submission.

**Additional Comments On Reviewer Discussion:**

The rebuttal provided some clarifications but did not adequately address the core concerns raised by reviewers, e.g., novelty of combining LLMs and EAs, reliance on manual initialization, limited scope of ablation studies, lack of detailed descriptions for key components.

---

### Decision · Program_Chairs · 2025-01-22

Reject